# Machine Teaching for Bayesian Learners in the Exponential Family

**Xiaojin Zhu**
Department of Computer Sciences, University of Wisconsin-Madison
Madison, WI, USA 53706
`jerryzhu@cs.wisc.edu`

## Abstract

What if there is a teacher who knows the learning goal and wants to design good training data for a machine learner? We propose an optimal teaching framework aimed at learners who employ Bayesian models. Our framework is expressed as an optimization problem over teaching examples that balance the future loss of the learner and the effort of the teacher. This optimization problem is in general hard. In the case where the learner employs conjugate exponential family models, we present an approximate algorithm for finding the optimal teaching set. Our algorithm optimizes the aggregate sufficient statistics, then unpacks them into actual teaching examples. We give several examples to illustrate our framework.

## 1 Introduction

Consider the simple task of learning a threshold classifier in 1D (Figure 1). There is an unknown threshold $\theta \in [0, 1]$. For any item $x \in [0, 1]$, its label $y$ is white if $x < \theta$ and black otherwise. After seeing $n$ training examples the learner's estimate is $\hat{\theta}$. What is the error $|\hat{\theta} - \theta|$? The answer depends on the learning paradigm. If the learner receives $iid$ noiseless training examples where $x \sim \mathrm{uniform}[0, 1]$, then with large probability $|\hat{\theta} - \theta| = O(\frac{1}{n})$. This is because the inner-most white and black items are $1/(n+1)$ apart on average. If the learner performs active learning and an oracle provides noiseless labels, then the error reduces faster $|\hat{\theta} - \theta| = O(\frac{1}{2^n})$ since the optimal strategy is binary search. However, a helpful teacher can simply teach with $n = 2$ items $(\theta - \epsilon/2, \mathrm{white}), (\theta + \epsilon/2, \mathrm{black})$ to achieve an arbitrarily small error $\epsilon$. The key difference is that an active learner still needs to explore the boundary, while a teacher can guide.

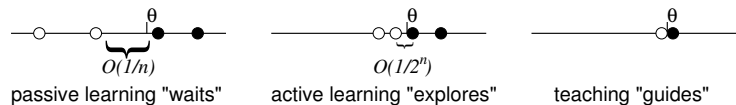

Figure 1: Teaching can require far fewer examples than passive or active learning

We impose the restriction that *teaching be conducted only via teaching examples* (rather than somehow directly giving the parameter $\theta$ to the learner). What, then, are the best teaching examples? Understanding the optimal teaching strategies is important for both machine learning and education: ($i$) When the learner is a human student (as modeled in cognitive psychology), optimal teaching theory can design the best lessons for education. ($ii$) In cyber-security the teacher may be an adversary attempting to mislead a machine learning system via "poisonous training examples." Optimal teaching quantifies the power and limits of such adversaries. ($iii$) Optimal teaching informs robots as to the best ways to utilize human teaching, and vice versa.

Our work builds upon three threads of research. The first thread is the teaching dimension theory by Goldman and Kearns [10] and its extensions in computer science(e.g., [1, 2, 11, 12, 14, 25]). Our framework allows for probabilistic, noisy learners with infinite hypothesis space, arbitrary loss functions, and the notion of teaching effort. Furthermore, in Section 3.2 we will show that the original teaching dimension is a special case of our framework. The second thread is the research on representativeness and pedagogy in cognitive science. Tenenbaum and Griffiths is the first to note that representative data is one that maximizes the posterior probability of the target model [22]. Their work on Gaussian distributions, and later work by Rafferty and Griffiths on multinomial distributions [19], find representative data by matching sufficient statistics. Our framework can be viewed as a generalization. Specifically, their work corresponds to the specific choice (to be defined in Section 2) of $loss()$ = KL divergence and $effort()$ being either zero or an indicator function to fix the data set size at $n$. We made it explicit that these functions can have other designs. Importantly, we also show that there are non-trivial interactions between loss() and effort(), such as not-teaching-at-all in Example 4, or non-brute-force-teaching in Example 5. An interesting variant studied in cognitive science is when the learner expects to be taught [20, 8]. We defer the discussion on this variant, known as "collusion" in computational teaching theory, and its connection to information theory to section 5. In addition, our optimal teaching framework may shed light on the optimality of different method of teaching humans [9, 13, 17, 18]. The third thread is the research on better ways to training machine learners such as curriculum learning or easy-to-hard ordering of training items [3, 15, 16], and optimal reward design in reinforcement learning [21]. Interactive systems have been built which employ or study teaching heuristics [4, 6]. Our framework provides a unifying optimization view that balances the future loss of the learner and the effort of the teacher.

## 2  Optimal Teaching for General Learners

We start with a general framework for teaching and gradually specialize the framework in later sections. Our framework consists of three entities: the world, the learner, and the teacher. ($i$) **The world** is defined by a target model $\theta^*$. Future test items for the learner will be drawn $iid$ from this model. This is the same as in standard machine learning. ($ii$) **The learner** has to learn $\theta^*$ from training data. Without loss of generality let $\theta^* \in \Theta$, the hypothesis space of the learner (if not, we can always admit approximation error and define $\theta^*$ to be the distribution in $\Theta$ closest to the world distribution). The learner is the same as in standard machine learning (learners who anticipate to be taught are discussed in section 5). The training data, however, is provided by a teacher. ($iii$) **The teacher** is the new entity in our framework. It is almost omnipotent: it knows the world $\theta^*$, the learner's hypothesis space $\Theta$, and importantly how the learner learns given any training data.[1] *However, it can only teach the learner by providing teaching (or, from the learner's perspective, training) examples.* The teacher's goal is to design a teaching set $\mathcal{D}$ so that the learner learns $\theta^*$ as accurately and effortlessly as possible. In this paper, we consider batch teaching where the teacher presents $\mathcal{D}$ to the learner all at once, and the teacher can use any item in the example domain.

Being completely general, we leave many details unspecified. For instance, the world's model can be supervised $p(x, y; \theta^*)$ or unsupervised $p(x; \theta^*)$; the learner may or may not be probabilistic; and when it is, $\Theta$ can be parametric or nonparametric. Nonetheless, we can already propose a generic optimization problem for optimal teaching:

$$\min_{\mathcal{D}} \quad \text{loss}(\widehat{f_{\mathcal{D}}}, \theta^*) + \text{effort}(\mathcal{D}). \tag{1}$$

The function loss() measures the learner's deviation from the desired $\theta^*$. The quantity $\widehat{f_{\mathcal{D}}}$ represents the state of the learner after seeing the teaching set $\mathcal{D}$. The function effort() measures the difficulty the teacher experiences when teaching with $\mathcal{D}$. Despite its appearance, the optimal teaching problem (1) is completely different from regularized parameter estimation in machine learning. The desired parameter $\theta^*$ is known to the teacher. The optimization is instead over the teaching set $\mathcal{D}$. This can be a difficult combinatorial problem – for instance we need to optimize over the cardinality of $\mathcal{D}$. Neither is the effort function a regularizer. The optimal teaching problem (1) so far is rather abstract. For the sake of concreteness we next focus on a rich family of learners, namely Bayesian models. However, we note that our framework can be adapted to other types of learners, as long as we know how they react to the teaching set $\mathcal{D}$.

# 3 Optimal Teaching for Bayesian Learners

We focus on Bayesian learners because they are widely used in both machine learning and cognitive science [7, 23, 24] and because of their predictability: they react to any teaching examples in $\mathcal{D}$ by performing Bayesian updates.[2] Before teaching, a Bayesian learner's state is captured by its prior distribution $p_0(\theta)$. Given $\mathcal{D}$, the learner's likelihood function is $p(\mathcal{D} \mid \theta)$. Both the prior and the likelihood are assumed to be known to the teacher. The learner's state after seeing $\mathcal{D}$ is the posterior distribution $\widehat{f_{\mathcal{D}}} \equiv p(\theta \mid \mathcal{D}) = \left( \int_{\Theta} p_0(\pi) p(\mathcal{D} \mid \pi) d\pi \right)^{-1} p_0(\theta) p(\mathcal{D} \mid \theta)$.

## 3.1 The KL Loss and Various Effort Functions, with Examples

The choice of $\mathrm{loss}()$ and $\mathrm{effort}()$ is problem-specific and depends on the teaching goal. In this paper, we will use the Kullback-Leibler divergence so that $\mathrm{loss}(\widehat{f_{\mathcal{D}}}, \theta^*) = KL\left(\delta_{\theta^*} \| p(\theta \mid \mathcal{D})\right)$, where $\delta_{\theta^*}$ is a point mass distribution at $\theta^*$.[3] This loss encourages the learner's posterior to concentrate around the world model $\theta^*$. With the KL loss, it is easy to verify that the optimal teaching problem (1) can be equivalently written as

$$\min_{\mathcal{D}} \quad -\log p(\theta^* \mid \mathcal{D}) + \mathrm{effort}(\mathcal{D}). \tag{2}$$

We remind the reader that this is not a MAP estimate problem. Instead, the intuition is to find a good teaching set $\mathcal{D}$ to make $\theta^*$ "stand out" in the posterior distribution.

The $\mathrm{effort}()$ function reflects resource constraints on the teacher and the learner: how hard is it to create the teaching examples, to deliver them to the learner, and to have the learner absorb them? For most of the paper we use the cardinality of the teaching set $\mathrm{effort}(\mathcal{D}) = c|\mathcal{D}|$ where $c$ is a positive per-item cost. This assumes that the teaching effort is proportional to the number of teaching items, which is reasonable in many problems. We will demonstrate a few other effort functions in the examples below.

How good is any teaching set $\mathcal{D}$? We hope $\mathcal{D}$ guides the learner's posterior toward the world's $\theta^*$, but we also hope $\mathcal{D}$ takes little effort to teach. The proper quality measure is the objective value (2) which balances the $\mathrm{loss}()$ and $\mathrm{effort}()$ terms.

**Definition 1** (Teaching Impedance). *The Teaching Impedance (TI) of a teaching set $\mathcal{D}$ is the objective value $-\log p(\theta^* \mid \mathcal{D}) + \mathrm{effort}(\mathcal{D})$. The lower the TI, the better.*

We now give examples to illustrate our optimal teaching framework for Bayesian learners.

**Example 1** (Teaching a 1D threshold classifier). *The classification task is the same as in Figure 1, with $x \in [0, 1]$ and $y \in \{-1, 1\}$. The parameter space is $\Theta = [0, 1]$. The world has a threshold $\theta^* \in \Theta$. Let the learner's prior be uniform $p_0(\theta) = 1$. The learner's likelihood function is $p(y = 1 \mid x, \theta) = 1$ if $x \geq \theta$ and $0$ otherwise.*

*The teacher wants the learner to arrive at a posterior $p(\theta \mid \mathcal{D})$ peaked at $\theta^*$ by designing a small $\mathcal{D}$. As discussed above, this can be formulated as (2) with the KL $\mathrm{loss}()$ and the cardinality $\mathrm{effort}()$ functions: $\min_{\mathcal{D}} \quad -\log p(\theta^* \mid \mathcal{D}) + c|\mathcal{D}|$. For any teaching set $\mathcal{D} = \{(x_1, y_1), \ldots, (x_n, y_n)\}$, the learner's posterior is simply $p(\theta \mid \mathcal{D}) = \mathrm{uniform}\left[\max_{i:y_i=-1}(x_i), \min_{i:y_i=1}(x_i)\right]$, namely uniform over the version space consistent with $\mathcal{D}$. The optimal teaching problem becomes $\min_{n, x_1, y_1, \ldots, x_n, y_n} \quad -\log\left(\frac{1}{\min_{i:y_i=1}(x_i) - \max_{i:y_i=-1}(x_i)}\right) + cn$. One solution is the limiting case with a teaching set of size two $\mathcal{D} = \{(\theta^* - \epsilon/2, -1), (\theta^* + \epsilon/2, 1)\}$ as $\epsilon \to 0$, since the Teaching Impedance $TI = \log(\epsilon) + 2c$ approaches $-\infty$. In other words, the teacher teaches by two examples arbitrarily close to, but on the opposite sides of, the decision boundary as in Figure 1(right).* □

**Example 2** (Learner cannot tell small differences apart). *Same as Example 1, but the learner has poor perception (e.g., children or robots) and cannot distinguish similar items very well. We may*

*encode this in* effort() *as, for example,* $\text{effort}(\mathcal{D}) = \frac{c}{\min_{x_i, x_j \in \mathcal{D}} |x_i - x_j|}$. *That is, the teaching examples require more effort to learn if any two items are too close. With two teaching examples as in Example 1, $TI = \log(\epsilon) + c/\epsilon$. It attains minimum at $\epsilon = c$. The optimal teaching set is $\mathcal{D} = \{(\theta^* - c/2, -1), (\theta^* + c/2, 1)\}$.* □

**Example 3** (Teaching to pick one model out of two). *There are two Gaussian distributions $\theta_A = N(-\frac{1}{4}, \frac{1}{2}), \theta_B = N(\frac{1}{4}, \frac{1}{2})$. The learner has $\Theta = \{\theta_A, \theta_B\}$, and we want to teach it the fact that the world is using $\theta^* = \theta_A$. Let the learner have equal prior $p_0(\theta_A) = p_0(\theta_B) = \frac{1}{2}$. The learner observes examples $x \in \mathbb{R}$, and its likelihood function is $p(x \mid \theta) = N(x \mid \theta)$. Let $\mathcal{D} = \{x_1, \ldots, x_n\}$. With these specific parameters, the KL loss can be shown to be $-\log p(\theta^* \mid \mathcal{D}) = \log\left(1 + \prod_{i=1}^{n} \exp(x_i)\right)$.*

*For this example, let us suppose that teaching with extreme item values is undesirable (note $x_i \to -\infty$ minimizes the KL loss). We combine cardinality and range preferences in $\text{effort}(\mathcal{D}) = cn + \sum_{i=1}^{n} \mathbb{I}(|x_i| \leq d)$, where the indicator function $\mathbb{I}(z) = 0$ if $z$ is true, and $+\infty$ otherwise. In other words, the teaching items must be in some interval $[-d, d]$. This leads to the optimal teaching problem $\min_{n, x_1, \ldots, x_n} \quad \log\left(1 + \prod_{i=1}^{n} \exp(x_i)\right) + cn + \sum_{i=1}^{n} \mathbb{I}(|x_i| \leq d)$. This is a mixed integer program (even harder–the number of variables has to be optimized as well). We first relax $n$ to real values. By inspection, the solution is to let all $x_i = -d$ and let $n$ minimize $TI = \log\left(1 + \exp(-dn)\right) + cn$. The minimum is achieved at $n = \frac{1}{d} \log\left(\frac{d}{c} - 1\right)$. We then round $n$ and force nonnegativity: $n = \max\left(0, \left[\frac{1}{d} \log\left(\frac{d}{c} - 1\right)\right]\right)$. This $\mathcal{D}$ is sensible: $\theta^* = \theta_A$ is the model on the left, and showing the learner $n$ copies of $-d$ lends the most support to that model. Note, however, that $n = 0$ for certain combinations of $c, d$ (e.g., when $c \geq d$): the effort of teaching outweighs the benefit. The teacher may choose to not teach at all and maintain the status quo (prior $p_0$) of the learner!* □

### 3.2 Teaching Dimension is a Special Case

In this section we provide a comparison to one of the most influential teaching models, namely the original teaching dimension theory [10]. It may seem that our optimal teaching setting (2) is more restrictive than theirs, since we make strong assumptions about the learner (that it is Bayesian, and the form of the prior and likelihood). Their query learning setting in fact makes equally strong assumptions, in that the learner updates its version space to be consistent with all teaching items. Indeed, we can cast their setting as a Bayesian learning problem, showing that their problem is a special case of (2). Corresponding to the concept class $C = \{c\}$ in [10], we define the conditional probability $P(y = 1 \mid x, \theta_c) = \begin{cases} 1, & \text{if } c(x) = + \\ 0, & \text{if } c(x) = - \end{cases}$ and the joint distribution $P(x, y \mid \theta_c) = P(x)P(y \mid x, \theta_c)$ where $P(x)$ is uniform over the domain $\mathcal{X}$. The world has $\theta^* = \theta_{c^*}$ corresponding to the target concept $c^* \in C$. The learner has $\Theta = \{\theta_c \mid c \in C\}$. The learner's prior is $p_0(\theta) = \text{uniform}(\Theta) = \frac{1}{|C|}$, and its likelihood function is $P(x, y \mid \theta_c)$. The learner's posterior after teaching with $\mathcal{D}$ is

$$P(\theta_c \mid \mathcal{D}) = \begin{cases} 1/(\text{number of concepts in } C \text{ consistent with } \mathcal{D}), & \text{if } c \text{ is consistent with } \mathcal{D} \\ 0, & \text{otherwise} \end{cases} \quad (3)$$

Teaching dimension $TD(c^*)$ is the minimum cardinality of $\mathcal{D}$ that uniquely identifies the target concept. We can formulate this using our optimal teaching framework

$$\min_{\mathcal{D}} -\log P(\theta_{c^*} \mid \mathcal{D}) + \gamma|\mathcal{D}|, \quad (4)$$

where we used the cardinality effort() function (and renamed the cost $\gamma$ for clarity). We can make sure that the loss term is minimized to 0, corresponding to successfully identifying the target concept, if $\gamma < \frac{1}{TD(c^*)}$. But since $TD(c^*)$ is unknown beforehand, we can set $\gamma \leq \frac{1}{|C|}$ since $|C| \geq TD(c^*)$ (one can at least eliminate one concept from the version space with each well-designed teaching item). The solution $\mathcal{D}$ to (4) is then a minimum teaching set for the target concept $c^*$, and $|\mathcal{D}| = TD(c^*)$.

## 4 Optimal Teaching for Bayesian Learners in the Exponential Family

While we have proposed an optimization-based framework for teaching any Bayesian learner and provided three examples, it is not clear if there is a unified approach to solve the optimization

problem (2). In this section, we further restrict ourselves to a subset of Bayesian learners whose prior and likelihood are in the exponential family and are conjugate. For this subset of Bayesian learners, finding the optimal teaching set $\mathcal{D}$ naturally decomposes into two steps: In the first step one solves a convex optimization problem to find the optimal aggregate sufficient statistics for $\mathcal{D}$. In the second step one "unpacks" the aggregate sufficient statistics into actual teaching examples. We present an approximate algorithm for doing so.

We recall that an exponential family distribution (see e.g. [5]) takes the form $p(x \mid \theta) = h(x) \exp\left(\theta^\top T(x) - A(\theta)\right)$ where $T(x) \in \mathbb{R}^D$ is the $D$-dimensional sufficient statistics of $x$, $\theta \in \mathbb{R}^D$ is the natural parameter, $A(\theta)$ is the log partition function, and $h(x)$ modifies the base measure. For a set $\mathcal{D} = \{x_1, \ldots, x_n\}$, the likelihood function under the exponential family takes a similar form $p(\mathcal{D} \mid \theta) = \left(\prod_{i=1}^n h(x_i)\right) \exp\left(\theta^\top \mathbf{s} - nA(\theta)\right)$, where we define

$$\mathbf{s} \equiv \sum_{i=1}^n T(x_i) \tag{5}$$

to be the aggregate sufficient statistics over $\mathcal{D}$. The corresponding conjugate prior is the exponential family distribution with natural parameters $(\lambda_1, \lambda_2) \in \mathbb{R}^D \times \mathbb{R}$: $p(\theta \mid \lambda_1, \lambda_2) = h_0(\theta) \exp\left(\lambda_1^\top \theta - \lambda_2 A(\theta) - A_0(\lambda_1, \lambda_2)\right)$. The posterior distribution is $p(\theta \mid \mathcal{D}, \lambda_1, \lambda_2) = h_0(\theta) \exp\left((\lambda_1 + \mathbf{s})^\top \theta - (\lambda_2 + n)A(\theta) - A_0(\lambda_1 + \mathbf{s}, \lambda_2 + n)\right)$. The posterior has the same form as the prior but with natural parameters $(\lambda_1 + \mathbf{s}, \lambda_2 + n)$. Note that the data $\mathcal{D}$ enters the posterior only via the aggregate sufficient statistics $\mathbf{s}$ and cardinality $n$. If we further assume that $\mathrm{effort}(\mathcal{D})$ can be expressed in $n$ and $\mathbf{s}$, then we can write our optimal teaching problem (2) as

$$\min_{n, \mathbf{s}} \ -\theta^{*\top}(\lambda_1 + \mathbf{s}) + A(\theta^*)(\lambda_2 + n) + A_0(\lambda_1 + \mathbf{s}, \lambda_2 + n) + \mathrm{effort}(n, \mathbf{s}), \tag{6}$$

where $n \in \mathbb{Z}_{\geq 0}$ and $\mathbf{s} \in \{t \in \mathbb{R}^D \mid \exists \{x_i\}_{i \in I} \text{ such that } t = \sum_{i \in I} T(x_i)\}$. We relax the problem to $n \in \mathbb{R}$ and $\mathbf{s} \in \mathbb{R}^D$, resulting in a lower bound of the original objective.[4] Since the log partition function $A_0()$ is convex in its parameters, we have a convex optimization problem (6) at hand if we design $\mathrm{effort}(n, \mathbf{s})$ to be convex, too. Therefore, the main advantage of using the exponential family distribution and conjugacy is this convex formulation, which we use to efficiently optimize over $n$ and $\mathbf{s}$. This forms the first step in finding $\mathcal{D}$.

However, we cannot directly teach with the aggregate sufficient statistics. We first turn $n$ back into an integer by $\max(0, [n])$ where $[]$ denotes rounding.[5] We then need to find $n$ teaching examples whose aggregate sufficient statistics is $\mathbf{s}$. The difficulty of this second "unpacking" step depends on the form of the sufficient statistics $T(x)$. For some exponential family distributions unpacking is trivial. For example, the exponential distribution has $T(x) = x$. Given $n$ and $\mathbf{s}$ we can easily unpack the teaching set $\mathcal{D} = \{x_1, \ldots, x_n\}$ by $x_1 = \ldots = x_n = \mathbf{s}/n$. The Poisson distribution has $T(x) = x$ as well, but the items $x$ need to be integers. This is still relatively easy to achieve by rounding $x_1, \ldots, x_n$ and making adjustments to make sure they still sum to $\mathbf{s}$. The univariate Gaussian distribution has $T(x) = (x, x^2)$ and unpacking is harder: given $n = 3, \mathbf{s} = (3, 5)$ it may not be immediately obvious that we can unpack into $\{x_1 = 0, x_2 = 1, x_3 = 2\}$ or even $\{x_1 = \frac{1}{2}, x_2 = \frac{5+\sqrt{13}}{4}, x_3 = \frac{5-\sqrt{13}}{4}\}$. Clearly, unpacking is not unique.

In this paper, we use an approximate unpacking algorithm. We initialize the $n$ teaching examples by $x_i \overset{iid}{\sim} p(x \mid \theta^*)$, $i = 1 \ldots n$. [6] We then improve the examples by solving an unconstrained optimization problem to match the examples' aggregate sufficient statistics to the given $\mathbf{s}$:

$$\min_{x_1, \ldots, x_n} \left\| \mathbf{s} - \sum_{i=1}^n T(x_i) \right\|^2. \tag{7}$$

This problem is non-convex in general but can be solved up to a local minimum. The gradient is $\frac{\partial}{\partial x_j} = -2\left(\mathbf{s} - \sum_i T(x_i)\right)^\top T'(x_j)$. Additional post-processing such as enforcing $x$ to be integers is then carried out if necessary. The complete algorithm is summarized in Algorithm 1.

---

**Algorithm 1** Approximately optimal teaching for Bayesian learners in the exponential family

---

**input** target $\theta^*$; learner information $T()$, $A()$, $A_0()$, $\lambda_1$, $\lambda_2$; effort()
    Step 1: Solve for aggregate sufficient statistics $n$, $\mathbf{s}$ by convex optimization (6)
    Step 2: Unpacking: $n \leftarrow \max(0, [n])$; find $x_1, \ldots, x_n$ by (7)
**output** $\mathcal{D} = \{x_1, \ldots, x_n\}$

---

We illustrate Algorithm 1 with several examples.

**Example 4** (Teaching the mean of a univariate Gaussian). *The world consists of a Gaussian $N(x; \mu^*, \sigma^2)$ where $\sigma^2$ is fixed and known to the learner while $\mu^*$ is to be taught. In exponential family form $p(x \mid \theta) = h(x)\exp(\theta T(x) - A(\theta))$ with $T(x) = x$ alone (since $\sigma^2$ is fixed), $\theta = \frac{\mu}{\sigma^2}$, $A(\theta) = \frac{\mu^2}{2\sigma^2} = \frac{\theta^2 \sigma^2}{2}$, and $h(x) = \left(\sqrt{2\pi}\sigma\right)^{-1}\exp\left(-\frac{x^2}{2\sigma^2}\right)$. Its conjugate prior (which is the learner's initial state) is Gaussian with the form $p(\theta \mid \lambda_1, \lambda_2) = h_0(\theta)\exp\left(\lambda_1\theta - \lambda_2\frac{\theta^2\sigma^2}{2} - A_0(\lambda)\right)$ where $A_0(\lambda_1, \lambda_2) = \frac{\lambda_1^2}{2\sigma^2\lambda_2} - \frac{1}{2}\log(\sigma^2\lambda_2)$.*

*To find a good teaching set $\mathcal{D}$, in step 1 we first find its optimal cardinality $n$ and aggregate sufficient statistics $s = \sum_{i\in\mathcal{D}} x_i$ using (6). The optimization problem becomes*

$$\min_{n,s} \quad -\theta^* s + \frac{\sigma^2\theta^{*2}}{2}n + \frac{(\lambda_1 + s)^2}{2\sigma^2(\lambda_2 + n)} - \frac{1}{2}\log(\sigma^2(\lambda_2 + n)) + \text{effort}(n, s) \tag{8}$$

*where $\theta^* = \mu^*/\sigma^2$. The result is more intuitive if we rewrite the conjugate prior in its standard form $\mu \sim N(\mu \mid \mu_0, \sigma_0^2)$ with the relation $\lambda_1 = \frac{\mu_0\sigma^2}{\sigma_0^2}$, $\lambda_2 = \frac{\sigma^2}{\sigma_0^2}$. With this notation, the optimal aggregate sufficient statistics is*

$$s = \frac{\sigma^2}{\sigma_0^2}(\mu^* - \mu_0) + \mu^* n. \tag{9}$$

*Note an interesting fact here: the average of teaching examples $\frac{s}{n}$ is not the target $\mu^*$, but should compensate for the learner's initial belief $\mu_0$. This is the "overshoot" discussed earlier. Putting (9) back in (8) the optimization over $n$ is $\min_n -\frac{1}{2}\log\sigma^2\left(\frac{\sigma^2}{\sigma_0^2} + n\right) + \text{effort}(n)$. Consider any differentiable effort function (w.r.t. the relaxed $n$) with derivative $\text{effort}'(n)$, the optimal $n$ is the solution to $n - \frac{1}{2\,\text{effort}'(n)} + \frac{\sigma^2}{\sigma_0^2} = 0$. For example, with the cardinality $\text{effort}(n) = cn$ we have $n = \frac{1}{2c} - \frac{\sigma^2}{\sigma_0^2}$.*

*In step 2 we unpack $n$ and $s$ into $\mathcal{D}$. We discretize $n$ by $\max(0, [n])$. Another interesting fact is that the optimal teaching strategy may be to not teach at all ($n = 0$). This is the case when the learner has literally a narrow mind to start with: $\sigma_0^2 < 2c\sigma^2$ (recall $\sigma_0^2$ is the learner's prior variance on the mean). Intuitively, the learner is too stubborn to change its prior belief by much, and such minuscule change does not justify the teaching effort.*

*Having picked $n$, unpacking $s$ is trivial since $T(x) = x$. For example, we can let $\mathcal{D}$ be $x_1 = \ldots = x_n = s/n$ as discussed earlier, without employing optimization (7). Yet another interesting fact is that such an alarming teaching set (with $n$ identical examples) is likely to contradict the world's model variance $\sigma^2$, but the discrepancy does not affect teaching because the learner fixes $\sigma^2$.*   □

**Example 5** (Teaching a multinomial distribution). *The world is a multinomial distribution $\pi^* = (\pi_1^*, \ldots, \pi_K^*)$ of dimension $K$. The learner starts with a conjugate Dirichlet prior $p(\pi \mid \beta) = \frac{\Gamma(\sum\beta_k)}{\prod\Gamma(\beta_k)}\prod_{k=1}^K \pi_k^{\beta_k - 1}$. Each teaching item is $x \in \{1, \ldots, K\}$. The teacher needs to decide the total number of teaching items $n$ and the split $\mathbf{s} = (s_1, \ldots, s_K)$ where $n = \sum_{k=1}^K s_k$.*

*In step 1, the sufficient statistics is $s_1, \ldots, s_{K-1}$ but for clarity we write (6) using $\mathbf{s}$ and standard parameters:*

$$\min_{\mathbf{s}} \quad -\log\Gamma\left(\sum_{k=1}^K (\beta_k + s_k)\right) + \sum_{k=1}^K \log\Gamma(\beta_k + s_k) - \sum_{k=1}^K (\beta_k + s_k - 1)\log\pi_k^* + \text{effort}(\mathbf{s}). \tag{10}$$

*This is an integer program; we relax $\mathbf{s} \in \mathbb{R}^K_{\geq 0}$, making it a continuous optimization problem with nonnegativity constraints. Assuming a differentiable* effort()*, the optimal aggregate sufficient statistics can be readily solved with the gradient $\frac{\partial}{\partial s_k} = -\psi\left(\sum_{k=1}^K (\beta_k + s_k)\right) + \psi(\beta_k + s_k) - \log \pi_k^* + \frac{\partial\text{effort}(\mathbf{s})}{\partial s_k}$, where $\psi()$ is the digamma function. In step 2, unpacking is again trivial: we simply let $s_k \leftarrow \lceil s_k \rceil$ for $k = 1 \dots K$.*

*Let us look at a concrete problem. Let the teaching target be $\pi^* = (\frac{1}{10}, \frac{3}{10}, \frac{6}{10})$. Let the learner's prior Dirichlet parameters be quite different: $\beta = (6, 3, 1)$. If we say that teaching requires no effort by setting* effort$(\mathbf{s}) = 0$*, then the optimal teaching set $\mathcal{D}$ found by Algorithm 1 is $\mathbf{s} = (317, 965, 1933)$ as implemented with Matlab* fmincon*. The MLE from $\mathcal{D}$ is $(0.099, 0.300, 0.601)$ and is very close to $\pi^*$. In fact, in our experiments,* fmincon *stopped because it exceeded the default function evaluation limit. Otherwise, the counts would grow even higher with MLE$\rightarrow \pi^*$. This is "brute-force teaching": using unlimited data to overwhelm the prior in the learner.*

*But if we say teaching is costly by setting* effort$(\mathbf{s}) = 0.3 \sum_{k=1}^K s_k$*, the optimal $\mathcal{D}$ found by Algorithm 1 is instead $\mathbf{s} = (0, 2, 8)$ with merely ten items. Note that it did not pick $(1, 3, 6)$ which also has ten items and whose MLE is $\pi^*$: this is again to compensate for the biased prior $\text{Dir}(\beta)$ in the learner. Our optimal teaching set $(0, 2, 8)$ has Teaching Impedance $TI = 2.65$. In contrast, the set $(1, 3, 6)$ has $TI = 4.51$ and the previous set $(317, 965, 1933)$ has $TI = 956.25$ due to its size. We can also attempt to sample teaching sets of size ten from* multinomial$(10, \pi^*)$*. In 100,000 simulations with such random teaching sets the average $TI = 4.97 \pm 1.88$ (standard deviation), minimum $TI = 2.65$, and maximum $TI = 18.7$. In summary, our optimal teaching set $(0, 2, 8)$ is very good.* □

We remark that one can teach complex models using simple ones as building blocks. For instance, with the machinery in Example 5 one can teach the learner a full generative model for a Naïve Bayes classifier. Let the target Naïve Bayes classifier have $K$ classes with class probability $p(y = k) = \pi_k^*$. Let $v$ be the vocabulary size. Let the target class conditional probability be $p(x = i \mid y = k) = \theta_{ki}^*$ for word type $i = 1 \dots v$ and label $k = 1 \dots K$. Then the aggregate sufficient statistics are $n_1 \dots n_K, m_{11} \dots m_{1v}, \dots, m_{K1} \dots m_{Kv}$ where $n_k$ is the number of documents with label $k$, and $m_{ki}$ is the number of times word $i$ appear in all documents with label $k$. The optimal choice of these $n$'s and $m$'s for teaching can be solved separately as in Example 5 as long as effort() can be separated. The unpacking step is easy: we know we need $n_k$ teaching documents with label $k$. These $n_k$ documents together need $m_{ki}$ counts of word type $i$. They can evenly split those counts. In the end, each teaching document with label $k$ will have the bag-of-words $\left(\frac{m_{k1}}{n_k}, \dots, \frac{m_{kv}}{n_k}\right)$, subject to rounding.

**Example 6** (Teaching a multivariate Gaussian). *Now we consider the general case of teaching both the mean and the covariance of a multivariate Gaussian. The world has the target $\mu^* \in \mathbb{R}^D$ and $\Sigma^* \in \mathbb{R}^{D \times D}$. The likelihood is $N(x \mid \mu, \Sigma)$. The learner starts with a Normal-Inverse-Wishart (NIW) conjugate prior $p(\mu, \Sigma \mid \mu_0, \kappa_0, \nu_0, \Lambda_0^{-1}) = \left(2^{\frac{\nu_0 D}{2}} \pi^{\frac{D(D-1)}{4}} \left(\prod_{i=1}^D \Gamma\left(\frac{\nu_0 + 1 - i}{2}\right)\right) |\Lambda_0|^{-\frac{\nu_0}{2}} \left(\frac{2\pi}{\kappa_0}\right)^{\frac{D}{2}}\right)^{-1} |\Sigma|^{-\frac{\nu_0 + D + 2}{2}}$ $\exp\left(-\frac{1}{2}\text{tr}(\Sigma^{-1}\Lambda_0) - \frac{\kappa_0}{2}(\mu - \mu_0)^\top \Sigma^{-1}(\mu - \mu_0)\right)$. Given data $x_1, \dots, x_n \in \mathbb{R}^D$, the aggregate sufficient statistics are $s = \sum_{i=1}^n x_i$, $\mathbb{S} = \sum_{i=1}^n x_i x_i^\top$. The posterior is NIW $p(\mu, \Sigma \mid \mu_n, \kappa_n, \nu_n, \Lambda_n^{-1})$ with parameters $\mu_n = \frac{\kappa_0}{\kappa_0 + n}\mu_0 + \frac{1}{\kappa_0 + n}s$, $\kappa_n = \kappa_0 + n$, $\nu_n = \nu_0 + n$, $\Lambda_n = \Lambda_0 + \mathbb{S} + \frac{\kappa_0 n}{\kappa_0 + n}\mu_0\mu_0^\top - \frac{2\kappa_0}{\kappa_0 + n}\mu_0 s^\top - \frac{1}{\kappa_0 + n}ss^\top$. We formulate the optimal aggregate sufficient statistics problem by putting the posterior into (6). Note $\mathbb{S}$ by definition needs to be positive semi-definite. In addition, with Cauchy-Schwarz inequality one can show that $\mathbb{S}_{ii} \geq s_i^2/2$ for $i = 1 \dots n$. Step 1 is thus the following SDP:*

$$\min_{n, s, \mathbb{S}} \quad \frac{D \log 2}{2}\nu_n + \sum_{i=1}^D \log \Gamma\left(\frac{\nu_n + 1 - i}{2}\right) - \frac{\nu_n}{2}\log|\Lambda_n| - \frac{D}{2}\log\kappa_n + \frac{\nu_n}{2}\log|\Sigma^*|$$

$$+ \frac{1}{2}\text{tr}(\Sigma^{*-1}\Lambda_n) + \frac{\kappa_n}{2}(\mu^* - \mu_n)^\top\Sigma^{*-1}(\mu^* - \mu_n) + \text{effort}(n, s, \mathbb{S}) \tag{11}$$

$$s.t. \quad \mathbb{S} \succeq 0; \quad \mathbb{S}_{ii} \geq s_i^2/2, \; \forall i. \tag{12}$$

*In step 2, we unpack $s, \mathbb{S}$ by initializing $x_1, \ldots, x_n \stackrel{iid}{\sim} N(\mu^*, \Sigma^*)$. Again, such iid samples are typically not good teaching examples. We improve them with the optimization (7) where $T(x)$ is the $(D + D^2)$-dim vector formed by the elements of $x$ and $xx^\top$, and similarly the aggregate sufficient statistics vector $\mathbf{s}$ is formed by the elements of $s$ and $\mathbb{S}$.*

*We illustrate the results on a concrete problem in $D = 3$. The target Gaussian is $\mu^* = (0, 0, 0)$ and $\Sigma^* = I$. The target mean is visualized in each plot of Figure 2 as a black dot. The learner's initial state is captured by the NIW with parameters $\mu_0 = (1, 1, 1), \kappa_0 = 1, \nu_0 = 2 + 10^{-5}, \Lambda_0 = 10^{-5}I$. Note the learner's prior mean $\mu_0$ is different than $\mu^*$, and is shown by the red dot in Figure 2. The red dot has a stem extending to the z-axis=0 plane for better visualization. We used an "expensive" effort function $\text{effort}(n, s, \mathbb{S}) = n$. Algorithm 1 decides to use $n = 4$ teaching examples with $s = (-1, -1, -1)$ and $\mathbb{S} = \begin{pmatrix} 4.63 & -1 & -1 \\ -1 & 4.63 & -1 \\ -1 & -1 & 4.63 \end{pmatrix}$. These unpack into $\mathcal{D} = \{x_1 \ldots x_4\}$, visualized by the four empty blue circles. The three panels of Figure 2 show unpacking results starting from different initial seeds sampled from $N(\mu^*, \Sigma^*)$. These teaching examples form a tetrahedron (edges added for clarity). This is sensible: in fact, one can show that the minimum teaching set for a D-dimensional Gaussian is the $D + 1$ points at the vertices of a D-dimensional tetrahedron. Importantly the mean of $\mathcal{D}$, $(-1/4, -1/4, -1/4)$ shown as the solid blue dot with a stem, is offset from the target $\mu^*$ and to the opposite side of the learner's prior $\mu_0$. This again shows that $\mathcal{D}$ compensates for the learner's prior. Our optimal teaching set $\mathcal{D}$ has $TI = 1.69$. In contrast, teaching sets with four iid random samples from the target $N(\mu^*, \Sigma^*)$ have worse TI. In 100,000 simulations such random teaching sets have average $TI = 9.06 \pm 3.34$, minimum $TI = 1.99$, and maximum $TI = 35.51$.* □

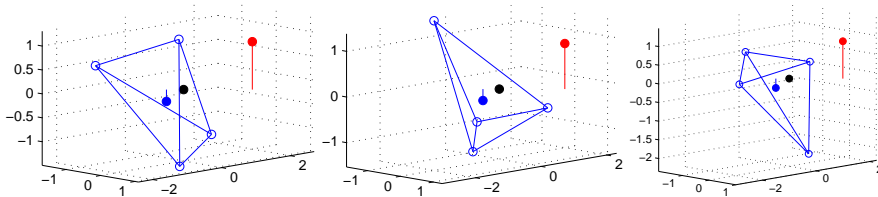

Figure 2: Teaching a multivariate Gaussian

## 5 Discussions and Conclusion

What if the learner *anticipates* teaching? Then the teaching set may be further reduced. For example, the task in Figure 1 may only require a single teaching example $\mathcal{D} = \{x_1 = \theta^*\}$, and the learner can figure out that this $x_1$ encodes the decision boundary. Smart learning behaviors similar to this have been observed in humans by Shafto and Goodman [20]. In fact, this is known as "collusion" in computational teaching theory (see e.g. [10]), and has strong connections to compression in information theory. In one extreme of collusion, the teacher and the learner agree upon an information-theoretical coding scheme beforehand. Then, the teaching set $\mathcal{D}$ is not used in a traditional machine learning training set sense, but rather as source coding. For example, $x_1$ itself would be a floating-point encoding of $\theta^*$ up to machine precision. In contrast, the present paper assumes that the learner does not collude.

We introduced an optimal teaching framework that balances teaching loss and effort. we hope this paper provides a "stepping stone" for follow-up work, such as 0-1 $\text{loss}()$ for classification, non-Bayesian learners, uncertainty in learner's state, and teaching materials beyond training items.

## Acknowledgments

We thank Bryan Gibson, Robert Nowak, Stephen Wright, Li Zhang, and the anonymous reviewers for suggestions that improved this paper. This research is supported in part by National Science Foundation grants IIS-0953219 and IIS-0916038.

## Footnotes

[1]This is a strong assumption. It can be relaxed in future work, where the teacher has to estimate the state of the learner by "probing" it with tests.

[2]Bayesian learners typically assume that the training data is *iid*; optimal teaching intentionally violates this assumption because the designed teaching examples in $\mathcal{D}$ will typically be non-*iid*. However, the learners are oblivious to this fact and will perform Bayesian update as usual.

[3]If we allow the teacher to be uncertain about the world $\theta^*$, we may encode the teacher's own belief as a distribution $p^*(\theta)$ and replace $\delta_{\theta^*}$ with $p^*(\theta)$.

[4] For higher solution quality we may impose certain convex constraints on $\mathbf{s}$ based on the structure of $T(x)$. For example, univariate Gaussian has $T(x) = (x, x^2)$. Let $\mathbf{s} = (s_1, s_2)$. It is easy to show that $\mathbf{s}$ must satisfy the constraint $s_2 \geq s_1^2/n$.

[5] Better results can be obtained by comparing the objective of (6) under several integers around $n$ and picking the smallest one.

[6] As we will see later, such $iid$ samples from the target distribution are not great teaching examples for two main reasons: (i) We really should compensate for the learner's prior by aiming not at the target distribution but overshooting a bit in the opposite direction of the prior. (ii) Randomness in the samples also prevents them from achieving the aggregate sufficient statistics.

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
