[Reviews · NeurIPS 2013]

Submitted by Assigned_Reviewer_5

This paper presents a detailed analysis of the problem of optimally teaching Bayesian learners, relating this analysis to previous work in learning theory. Results are presented for optimal teaching for exponential family distributions, using the Gaussian and multinomial cases as examples. The optimal strategy is essentially selecting a adataset that has the correct sufficient statistics.

The analysis is nice and the paper is generally clear. However, it fails to make contact with a related literature in cognitive science that anticipates many of the theoretical results. The same solution to the problem of providing "representative" data is presented in Tenenbaum & Griffiths (2001), including an analysis of the Gaussian case that produces the same conclusions. Extensions of this idea to the multinomial distribution appear in Rafferty & Griffiths (2010), making the same point about matching sufficient statistics. This work should minimally be acknowledged, and the connections identified. I'm not sure how much the present contribution goes beyond this previous work.

In addition, research in cognitive science has looked at a more complex version of this problem: what happens when the learner also knows that the teacher is providing them with good examples. Shafto & Goodman (2008) built on the framework introduced by Tenenbaum & Griffiths (2001) to address this question, which can result in significantly different answers from the simple idea of matching sufficient statistics. In the context of the classification example presented in the paper, just one example is sufficient to convey the boundary if the learner also knows that they are being taught, since the most efficient way to communicate the boundary is to put a labeled datapoint epsilon units to one side of it.

References

Rafferty, A. N., & Griffiths, T. L. (2010). Optimal language learning: The importance of starting representative. Proceedings of the 32nd Annual Conference of the Cognitive Science Society.

Shafto, P. & Goodman, N. (2008). Teaching games: Statistical sampling assumptions for learning in pedagogical situations. Proceedings of the 30th annual conference of the Cognitive Science Society.

Tenenbaum, J. B., & Griffiths, T. L. (2001). The rational basis of representativeness. Proceedings of the 23rd Annual Conference of the Cognitive Science Society.
Summary: This is nice work, although the extent of its contribution relative to previous work in cognitive science is not clear.

Submitted by Assigned_Reviewer_6

The paper presents a framework for machine teaching. A general framework is presented, which is then restricted to Bayesian learners. Several examples are provided in which the optimal teaching examples are derived. Finally, a framework for exponential family distributions is presented.

In general, I find the work very interesting. There are several aspects that are not as strong as they could be. First, the degree of novelty of the framework is debatable. There has been considerable work on very similar frameworks in cognitive science including Griffiths & Tenenbaum, 2001; Shafto & Goodman, 2008; Goodman & Frank, 2012. All of these formalize models of teaching (or something very similar) based on the idea of providing examples to Bayesian learners. It is surprising that none of this work is cited, given the relevance to the current paper.

Second, the first set of three examples are of limited interest (learning a threshold classifier, learning a classifier via noisy observation, and learning which of two Gaussians produced some data.

The section on the exponential family is the most interesting and novel. Given the general formalization of the problem, though the specific examples are rather long.

Finally, the title far outpaces the content of the paper. It is very interesting to have a framework for teaching exponential family distributions, but that does not encompass all of what might be called “Bayesian learners”.
Summary: Very promising work, but not quite ready for NIPS.

Submitted by Assigned_Reviewer_7

This work is really cool, and would make a fine addition to NIPS. The paper is
beautifully written and accessible even to a nontheorist like myself.
Developing a formal theory of teaching seems like a very important long-term
objective. However, I'm skeptical that the research presented in this
manuscript -- and any other research examining classification learning purely
from examples with no additional communication from a teacher -- will have
much practical impact. Hopefully in the long run, the work can be expanded in
the direction of more naturalistic assumptions.

The work has two other strong limitations.

(1) Human students are not Bayesian learners. Consequently, good teachers use
feedback from the student to determine the student's misconceptions
(deviations from Bayesian learning?) and to adapt the sequence of examples.
With the assumption of the present work that the learner treats training
examples as iid, the sequencing of examples is irrelevant, as is the feedback
from the student, as is the need for the teacher to adapt examples based on
the feedback.

(2) In a human teaching scenario, typically the learner's prior beliefs are
not known, whereas in the present work, these priors are given. The key
simulation results in the manuscript follow from the availability of the
learner's priors; these results show that appropriate choice of training
examples can overcome prior beliefs.

I wasn't convinced that the optimal (n,s) found via convex optimization can be
transformed into a set of teaching examples that match the sufficient
statistics to a given degree of accuracy. What assurance is there that the
sufficient statistics can be captured by ceil(n) examples? The larger
question is: what assurance do we have that the teaching problem can be
decomposed into (a) the determination of sufficiency statistics and 'n', and
(b) the determination of examples from the statistics.
Summary: The work tackles an important problem from a novel theoretical perspective, and takes a clear step toward solving the problem. The work should be of interest to theorists as well as cognitive scientists.

Submitted by Assigned_Reviewer_8

The authors generalize the teaching dimension theory [Goldman & Kearns, 95] to the family of Bayesian Learners (and hypothetically to noisy teachers) -- although they restrict any actual analysis to exponential families of distributions (or at least distributions with conjugate priors -- which admittedly much of machine learning does). The basic model minimizes the combination of loss from the resulting learner hypothesis and effort of the teacher. Based of the basic idea of selecting a dataset that has the correct sufficient statistics, they first provide a solution to the 1D threshold function (under two different scenarios), a restricted model selection problem (one of two Gaussian distributions), and eventually generalize this to teaching Bayesian Learners of the family of Exponential Distributions.

Overall, the paper is very clear, although quite a bit of space is spent on trivial examples while the exponential family section is by far the most interesting and practical. My biggest concern with this paper is that while it does improve the teaching dimension literature, it seems to think it is making a bigger jump than it is (and sort of overselling its claims a bit to be honest) as it is missing some reference to statistics/cognitive science literature that have very similar results (I am not even sure if “mostly studied from th query learning angle” is entirely correct -- although it certainly has been studied from this angle). For example, [Tenenbaum & Griffiths, 2001; Shafto & Goodman, 2008; Rafferty & Griffiths, 2010]. There are clearly some very strong connections here and the work should at least be acknowledged (and I actually think could save some space for discussion on the exponential families model) -- although I do admit that I like the presentation better in this paper (but this is likely since I am not a cognitive scientist).

While I am not a cognitive scientist, I am also somewhat concerned about the claims regarding the application to human learning. Given the authors knowledge of query learning, I am actually somewhat surprised by this claim. My understanding is that the current theory is that humans are not passive Bayesian learners -- not only is the presentation of the examples by the teacher not iid, but it is influenced by feedback from the students (which I suppose could hypothetically be modeled as some sampling from the distribution of the student) and if nothing else, the prior of the student are certainly not known (although, again, I suppose they could be probed).

That being said, I think it is a nice work overall and would be interesting to the NIPS community (in particular relative to the machine learning community in general). It is a very clean presentation, made me think a bit, and would likely influence my work -- which means it would likely be of interest to the NIPS community (even if I think it is a bit “in progress” with respect to contextualizing within related work).
Summary: This is a nice work overall, which I found more interesting than most NIPS submissions. The biggest weaknesses are contextualizing wrt some very related work and overstating the potential impact on human learning models (in my opinion).
Author Feedback

Author rebuttal: Dear reviewers: Many of you pointed out that we failed to contextualize out work with respect to those in cognitive science. We agree. It was our negligence, and will be corrected in the final version by placing our work in the continuum of those pioneering papers and tuning down our claims accordingly. Thank you.

In response to your questions on significance/novelty, ours is a non-trivial generalization of those work:

First, the representativeness framework in Tenenbaum & Griffiths 2001 and later papers is equivalent to the specific choice of loss()=KL, and effort() either non-existent or fixing a training set size at n. We made it explicit that these functions can have other designs. Importantly, we also showed that there are non-trivial interactions between loss() and effort(), such as not-teaching-at-all in Example 4, or non-brute-force-teaching in Example 5.

Second, they worked out specific models such as Gaussian and multinomial. Our algorithm 1 provides a generic method for conjugate models.

Finally, by abstracting out loss() and effort() we hope this paper provides a "step stone" in anticipation of follow up work, such as 0-1 loss() for classification, uncertainty in learner's prior, or non-Bayesian learners.

Reviewer 5: Thank you for pointing out Shafto & Goodman 2008 where the learner knows the teacher is teaching. This is extremely interesting, since it corresponds to a setting that some computational teaching theory papers actually tried to "avoid." It was called collusion. Despite the negative sentiment, the math is elegant: if the learner and the teacher collude to the full extent, they essentially agree on a common and efficient encoding of the models. Then, teaching examples are not used in a traditional machine learning training set sense, but rather as source coding as per information theory. In your 1D example, that single labeled data point is indeed optimal; it is a k-bit encoding of the decision boundary up to that resolution. This connection is worth adding to the paper.